# Leaving no one behind in health: Financial hardship to access health care in Ethiopia

**Yawkal Tsega**[1]*, **Gebeyehu Tsega**[2], **Getasew Taddesse**[2], **Gebremariam Getaneh**[2]

**1** School of Public Health, College of Medicine and Health Sciences, Wollo University, Dessie, Ethiopia,
**2** Department of Health Systems Management and Health Economics, School of Public Health, College of
Medicine and Health Sciences, Bahir Dar University, Bahir Dar, Ethiopia

* yawkaltsega@gmail.com

**Data Availability Statement:** All relevant data were included in the paper and Supporting Information files.

**Funding:** The author(s) received no specific funding for this work. We (all authors) have

## Abstract

### Background

Financial hardship (of health care) is a global and a national priority area. All people should be protected from financial hardship to ensure inclusive better health outcome. However, financial hardship of healthcare has not been well studied in Ethiopia in general and in Debre Tabor town in particular. Therefore, this study aimed to assess the incidence of financial hardship of healthcare and associated factors among households in Debre Tabor town.

### Methods

Community based cross sectional study was conducted, from May 24/2022 to June 17/2022, on 423 (selected through simple random sampling) households. Financial hardship was measured through catastrophic (using 10% threshold level) and impoverishing (using $1.90 poverty line) health expenditures. Patient perspective bottom up and prevalence based costing approach were used. Indirect cost was estimated through human capital approach. Bi-variable and multiple logistic regressions were used.

### Results

The response rate was 95%. The mean household annual healthcare expenditure was Ethiopian birr 12050.64 ($227.37). About 37.1% (95%CI: 32, 42%) of the households spend catastrophic health expenditure with a 10% threshold level and 10.4% of households were impoverished with $1.90 per day poverty line. Being old, with age above 60, (AOR: 4.21, CI: 1.23, 14.45), being non-insured (AOR: 2.19, CI: 1.04, 4.62), chronically ill (AOR: 7.20, CI: 3.64, 14.26), seeking traditional healthcare (AOR: 2.63, CI: 1.37. 5.05) and being socially unsupported (AOR: 2.77, CI: 1.25, 6.17) were statistically significant factors for catastrophic health expenditure.

### Conclusion

The study showed that significant number of households was not yet protected from financial hardship of healthcare. The financial hardship of health care is stronger among the less privileged populations: non-insured, the chronically diseased, the elder and socially

reviewed the manuscript and agreed to submit to your journal, PLOS ONE.

**Competing interests:** The authors have declared that no competing interests exist.

unsupported. Therefore, financial risk protection strategies should be strengthened by the concerned bodies.

## Introduction

Universal health coverage (UHC), one (the overarching) target of Sustainable Development Goals (SDGs), ensures that all people receive quality essential health services they need without exposing them to financial hardship. Financial risk protection is at the core of universal health coverage and it is one priority area in Ethiopian health sector as indicated in Health Sector Transformational Plan two (HSTP II). It is achieved when there are no financial barriers (mainly due to direct out of pocket health expenditure) to access essential health services [1–3]. Out of pocket (OOP) health spending is defined as any spending incurred by a household when any member uses a healthcare, including promotive, preventive, curative, rehabilitative and palliative care. To access (high quality) health care, the household incurs direct medical and non-medical costs, indirect cost and intangible cost. These costs impose financial hardship to the households, and worst in low income countries like Ethiopia [1,2].

Financial hardship (FH) is measured through Catastrophic Health Expenditure (CHE) and Impoverishing Health Expenditure (IHE). These metrics are standards that used to monitor and track Sustainable Development Goal indicator 3.8.2 (SDG indicator 3.8.2) across United Nations (UN) member states. CHE is considered when healthcare spending exceeds a certain threshold (varied from 10% to 40%) of household expenditure or income. From these thresholds, 10%(the lower threshold level) and 25%(the higher threshold level) are used in a joint report of World Bank(WB) and World Health Organization(WHO), a report in every 2 years, for monitoring and tracking SDG indicator 3.8.2. Whereas, IHE is considered when households' health expenditure is making the households below a given poverty line (in our cases a World Bank $1.9 a day extreme poverty line) or further impoverish to extreme poverty [1,2,4].

Globally, the incidence of financial hardship of healthcare has been increased since 2000. For example, the incidence of CHE increased by 3.6% annually, from 571 million in 2000 to 927 million in 2015 with 10% threshold level. Similarly, the incidence of catastrophic health expenditure has increased from 12.7% in 2015 to 13.2% in 2017 at 10% threshold level. CHE, as measured by SDG indicator 3.8.2, will continue to rise to the year 2030 if the share of out-of-pocket health spending continues at its current rate [1,2].

Furthermore, OOP healthcare costs lead more people falling into poverty. About 89.7 million individuals (1.2% of global population) were forced into extreme poverty (below $1.90 a day poverty line) and 98.8 million (1.4% of global population) were pushed below $3.20 a day poverty line and 183.2 million were pushed into poverty defined in relative terms (below 60% of median daily per capita consumption or income in their country). At all of these poverty levels, lower and middle-income countries (LMICs) had the highest number and proportion of the world's population with impoverishment due to OOP health spending [1,2].

These financial burden (CHE&IHE) contributes to socioeconomic disparities in access to essential healthcare services. The burden is directly proportional to the severity of the underlying health condition (ill individuals spend more). Households seeking care face barriers to access essential health services related to financial hardships. This leads to people delayed or forgone essential health services [2,5,6].

In the majority of LMICs, low health care resources and a lack of protection from catastrophic healthcare costs have led to an over-reliance on OOP health spending. Households who are dependent on OOP healthcare payment and who are unable to cope with the

economic implications of illness are frequently pushed into poverty. Households in this scenario incur more financial obligations and lack the resources to meet other basic requirements such as food and education [7].

Moreover, in low-income countries, OOP health expenditures accounts for more than half of overall spending and more than one third in middle-income countries. According to World Health Organization (WHO), OOP payments push millions of households into absolute poverty each year, and many of them are at risk of catastrophic health expenditure since their OOP healthcare expenses are equivalent to or exceed 40% of their income or expenditure. Many families forego services because of the direct and indirect health expenditures exceed their financial means. Because of the loss of income caused by disease, poor households become increasingly poorer, and overall quality of life suffers even more [8].

Catastrophic health payments are concentrated among the poor, including African countries. Inequities in access exist in Sub-Saharan African (SSA) countries as a result of income disparities and the level of OOP health expenditure within the country. The percentage of households suffering by catastrophic health care expenses has been proven to differ significantly among countries [9].

Since financial hardship of health care is a main challenge and a priority area of the health sector, Ethiopian healthcare financing reform has been implemented before 24 years, since 1998. For example, various financial hardship protection measures like fee waiver system, exempted services(e.g. maternal health services) and community based health insurance have been implemented in Ethiopia [10]. However, OOP health expenditures continue to be a considerable financial burden of households. For example, as per the latest national health account, the seventh Ethiopian Health Account (EHA), OOP health spending amounted to 31% of the total health expenditure, which is unacceptably high and it is higher than that of the global recommended target, 20% [11,12]. As a result, households often obliged to borrow money, sell their assets, reduce consumption of other basic needs to spend on healthcare expenditure and my forgone the healthcare services [13,14].

Our literature review indicted that financial hardship of health care can be affected by several factors(11, 12). Based on the review, we developed a conceptual framework (Fig 1) that depicts the potential relationships between outcome (financial hardship of healthcare) and explanatory variables.

Evidence, on the magnitude of financial hardship of healthcare and its determinant factors at household level, is critical to ensure effective, equitable and affordable access to quality health services that will achieve the motto of "leave no one behind" as stated in SDG 3.8.2 and HSTP II. However, it has not been well studied in Ethiopia in general and in Debre Tabor town in particular. Therefore, the aim of this study is to assess the financial hardship of healthcare and its associated factors among households in Debre Tabor town, South Gondar zone, Ethiopia.

## Methods and materials

### Study design and period

Community based cross-sectional study design was conducted to assess financial hardship of healthcare and associated factors among households in Debre Tabor town from May 24/2022 to June 17/2022.

### Study area and setting

The study was conducted in Debre Tabor town, Amhara regional state of Ethiopia. Debre Tabor town is the capital of South Gondar Zone and has six kebeles with 19,624 households.

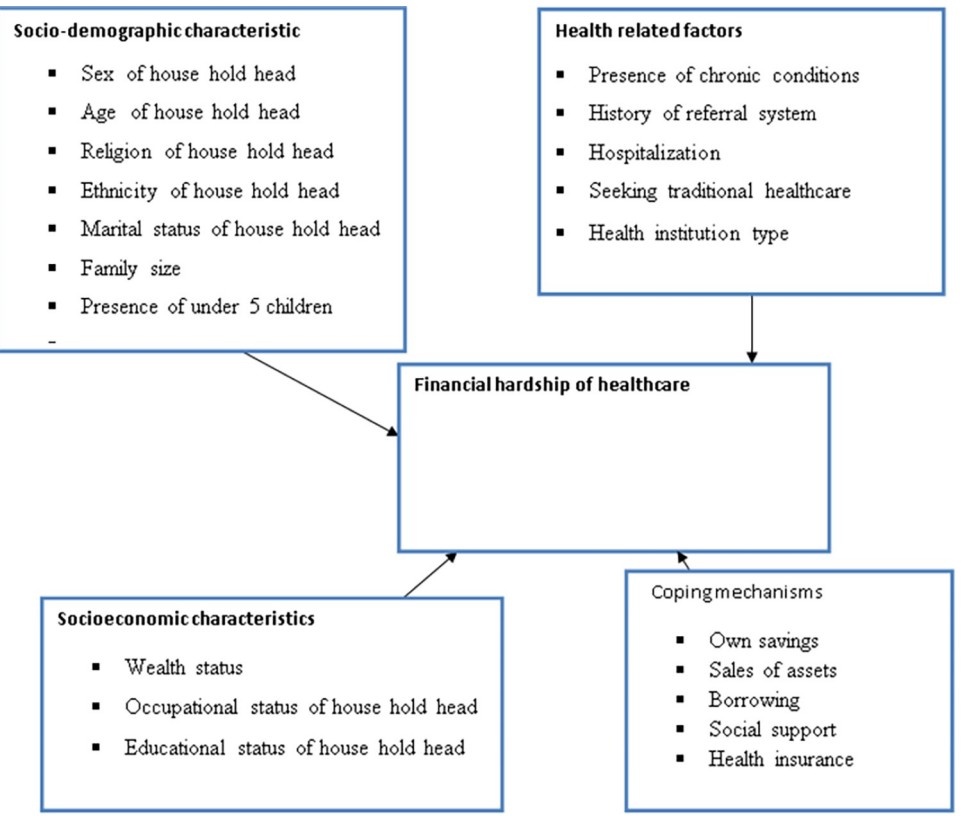

**Fig 1. Conceptual framework depicts relationships between financial hardship of health care and predictor variables.**

The town has 84,382 populations of which 19,898 are in reproductive age group and 10,868 are children from age 6 to 59 months. The town is located at 108.6 kilo meters east of capital of Amhara state, Bahir Dar city. The town has one public hospital namely Debre Tabor comprehensive specialized hospital and three health centers namely Leul Alemayehu, Tabor and Debre Tabor health centers [15].

**Study population.**   All households in Debre Tabor town were the study populations.

**Sampling unit.**   The sampling unit of the study was households

**Study unit.**   The study unit of this study was household heads

**Inclusion criteria.**   All households lived in Debre Tabor town for 6 months and above were included in the study.

**Exclusion criteria.**   Household heads unable to respond due to different reasons were excluded from the study.

## Dependent variable

✓ Catastrophic health expenditure

### Independent variables.

✓ Socio demographic factors (sex of household head, age of household head, religion of household head, ethnicity, marital status of household head and family size)

✓ Socioeconomic factors (wealth status, presence of under five children, educational status, and occupational status)

✓ Health related variables (presence of chronic health conditions, history of referral, traditional healthcare seeking, health institution type, hospitalization)

✓ Coping strategies (insurance status, selling household assets, own saving, borrowing and main source of fund for health cost from social support)

**Operational and term definitions.** Financial hardship of healthcare: defined as a situation where the household is having difficulty to pay health care cost. It is measured through CHE and IHE

Adult equivalent: All members of the household with adjusted calorie need requirement on the basis of age and sex [16].

Catastrophic health expenditure (CHE)**:** spending greater than 10% of household's reported total expenditure for healthcare service [1,2].

Poverty line (PL): WB poverty line ($1.90 a day extreme poverty line)was used in this study [1,2].

Healthcare expenditure: The total household expenditure related to healthcare which included direct medical, direct non-medical and indirect costs [1].

Impoverishing health expenditure (IHE): When households pushed below $1.90 a day extreme poverty line because of their healthcare expenditure, it was considered as IHE [1].

Poverty gap (poverty gap index): How far households are from the poverty line (measures intensity of poverty) [17].

Wealth index: The composite measure of cumulative living standard of the household. It was measured by 35 items [12,18].

Chronic health condition: is a human health condition or disease that is persistent or otherwise long-lasting in its effects or a disease that comes with time [19]. The term chronic was used when the course of the disease lasts for more than three months Health insurance: in this study means community based health insurance(CBHI) [20].

Healthcare cost measurement

**Types of costs and their costing methods.** There are two methods of costing approaches such as the prevalence and incidence approaches. The prevalence method is the commonest costing approach in studies and was used in this study [21,22].

We estimated the direct medical and nonmedical costs, and indirect costs. Since, intangible costs are difficult to measure, we did not measure the intangible costs. The direct medical costs included costs of registration cards, medications, imaging diagnostic tests, laboratory and bed incurred 12 months back the study conducted and direct non-medical costs include cost of transportation, cafeteria and lodging while seeking healthcare service both for the patient and the caregiver [23–25]. Bottom up (micro) costing approach was used based on average cost of health care services to estimate the direct costs of healthcare services [26,27].

Moreover, annual average expenditure on healthcare for each household was estimated by summing up all self-reported healthcare expenditures from May 2021 to May 24/2022. Similarly, all the expenditures for transportation, cafeteria and lodging was summed up based on the self-reported number of household members having history of illness and amount of money they incurred.

Data on indirect costs covered in this study included lost days (absenteeism) both for the patient and caregiver as per human capital approach.

For workers(payroll paid and merchants), monetary value of lost days was calculated by multiplying number of lost days with reported personal daily income (monthly income

divided by 30). For non-payroll paid households, their reported annual household income was used.

**Measurement of catastrophic and impoverishing health expenditure.** Wagstaff and van Doorslaer approach was used to measure CHE and IHE. This approach considers catastrophic health expenditure when the proportion of household's health expenditure as a share of total household expenditure/income or nonfood expenditure exceeded a specific threshold level. The choice regarding the threshold to use in determining catastrophic health expenditure is arbitrary and has typically varied from 10% to 40% [28].

To calculate the catastrophic head count which is the percentage of households incurring catastrophic expenditures, we defined $T_{HE}$ as total annual health expenditures for household i, $T_E$ total annual expenditure for household i, and $F_E$ for food expenditures for household i.

A household was considered to have catastrophic health expenditure if $T_{HE}/T_E$ surpassed a specified threshold, Z (in our case 10% threshold level was used).

The catastrophic headcount (Hc) is the given by:-

$$Hc = \frac{1}{N}\sum_{i=1}^{N} E_i \tag{1}$$

Where N is the sample size and $E_i$ equals 1 if $T_{HE}/T_E > z$ and zero otherwise.

The headcount does not reflect the amount by which households exceed the threshold. Therefore, we used the catastrophic expenditure overshoot which captures the average degree by which health expenditures (as a proportion of total expenditure or non-food expenditure) exceed the threshold z. The overall overshoot (O) is given by:-

$$O = \frac{1}{N}\sum_{i=1}^{N} O_i \tag{2}$$

Where $O_i = E_i ((T_{HE}/T_E) - z)$.

Where $E_i = ((T_{HE}/T_E)-z)$ if $(T_{HE}/T_E)>z$, and 0 otherwise.

The incidence (headcount) and the intensity (overshoot) of catastrophic expenditures are related through the mean positive overshoot (MPO) which captures the intensity of occurrence of catastrophic expenditures defined as overshoot divided by headcount:

$$MPO = \frac{O}{H}; \ O = H^*MPO \tag{3}$$

Wagstaff and van Doorslaer also describe methods to adjust poverty measures on the basis of household expenditure net of OOP spending on health care(28). The three measures of poverty include;

1).Poverty head count, which is the proportion of households living below the poverty line ($1.9 a day extreme poverty line);

$$H_{pov}^{pre} = 1/N\sum_{i=1}^{N} P_i^{pre} = \mu Ppre \tag{4}$$

Where $H_{pov}^{pre}$ is poverty headcount before health payment and $P_i pre = 1$ if $X_i < PL$ and zero otherwise.

2). Poverty gap, referring to the aggregate of all short falls from the poverty line;

$$G_{pov}^{pre} = 1/N \sum_{i=1}^{N} g_i^{pre} = {}^{\mu} gpre \tag{5}$$

Where $G_{pov}^{pre}$ is prepayment poverty gap, $g_i^{pre}$ = PL-X$_i$ if PL>X$_i$ and zero otherwise.

3). Normalized poverty gap ($N\,G_{pov}^{pre}$) or poverty gap index is obtained by dividing the poverty gap by the poverty line.

$$NG_{pov}^{pre} = \frac{G_{pov}^{pre}}{PL} \tag{6}$$

Calculating the three measures requires setting a poverty line and assessing the extent to which health care payments push households below the poverty line. The World Bank poverty line 1.9 US dollar per person per day was converted to ETB based on average exchange rate (1USD = ETB 53) of September 2021 to August 2022 was used to estimate poverty levels before and after healthcare expenditure. Replacing all the pre-payment superscripts, 'pre' by the superscript 'post' gives the analogous post-payment poverty measurement.

The measures of poverty impact (PIH) of health expenditure are then simply defined as the difference between the pre-payment and post-payment measures, i.e.

$$PI^{H} = H_{pov}^{post} - H_{pov}^{pre} \tag{7}$$

## Sample size determination and sampling methods

**Sample size determination.** Single population proportion formula was used to estimate the sample size, by taking the proportion 50% of CHE at 10% threshold level with confidence level of 95% and degree of precision 5% and non-response rate of 10% is considered and then the total sample size was;

$$n = \frac{Z(\alpha/2)2 * P(1-q)}{d^2}$$

Where P = 50%
d = 0.05 (degree of precision) and Z $\alpha/2$ at 95% confidence level = 1.96
By taking the above values, the sample size was

$$n = \frac{(1.96)^2 * (0.5)(1 - 0.5)}{(0.05)^2} = 384$$

Non-response rate (NRR) = 10%; Therefore; the number of households included in the study were 384*10%NRR + 384 = 423.

**Sampling method and procedures.** Computer generated simple random sampling method was used. The list of eligible households was obtained from urban health extension professionals and used as a sampling frame. Households were listed and coded (from 1 to 19,624). Then, households were selected using OpenEpi application computer generated simple random sampling method.Abrahajira Hospital **72(20)** employees

## Survey instruments and data collection procedures

Structured questionnaire was used. The questionnaire was developed after reviewing various literatures. The survey instrument included categories aim to collect data on

sociodemographic and socioeconomic characteristics, health profile and related characteristics of households, total expenditures of the household, total health expenditures and coping mechanisms of catastrophic health expenditure (S1 Text). A pretested and interviewer administered questionnaire was used. Two data collectors who have bachelor of degree (public health graduates) and one supervisor (MPH) were employed. Total annual (from May 2021 to May 24/2022) health care and other household expenditures were collected from the head of each selected household. Each healthcare, food and non-food expenditures were summed up and total annual health expenditure, total annual food expenditure, total annual nonfood expenditure and total annual household expenditure, which used as a denominator to calculate catastrophic health expenditure, were determined. Furthermore, the wealth index assessing variables were adapted from Ethiopian DHS 2019 for urban area. About 35 questions assessing sanitation facility, drinking water source, housing condition and ownership of durable assets were asked to the household head.

**Data management and analysis.** The collected data were checked for completeness. Then, data were coded, organized and entered into EpiData version 3.1 and exported to SPSS version 25 for analysis. Descriptive statistical analysis (frequencies and percent), bi-variable and multiple logistic regressions were conducted. In bi-variable logistic regression, variables having P-value of $<0.2$ with 95% confidence interval were eligible to multiple logistic regression. The overall goodness fit of binary logistic regression model was checked by Hosmer and Lemeshow test (p-value $>0.05$). Assumptions of binary logistic regression such as multicollinearity and outliers were checked for the model. Adjusted Odds Ratio (AOR) with 95% confidence intervals was estimated to assess the strength of the association, and a p-value of $< 0.05$ was used to declare statistically significant factors.

Wealth index was constructed using principal component analysis by SPSS. Wealth index construction question scores was derived using principal component analysis in that; 35 wealth status assessing variables from sanitation, housing condition, water source and household durable assets was computed. Variables having frequency of greater than 95% and less than 5% were excluded. In principal component analysis output of correlation matrix, values less than 0.1 and greater than 0.9 were removed from the analysis. After all, 12 variables were used to construct wealth index. The first component of the composite variables was used to estimate wealth status of households and ranked in ascending order.

**Data quality assurance.** The structured questionnaires were prepared in English first and translated to Amharic with clear way for better understanding with respondents. Two data collectors with educational level of bachelor of degree (public health graduates) and one supervisor (MPH) were employed. Three days training was given for data collectors on the overall picture of questionnaires, how to collect the data and how to approach the respondents. Before actual data collection, pretesting on 5% of the sample size was done at Woreta town. Close supervision of data collectors was done and data were checked for its completeness on daily basis.

## Ethical consideration

Ethical clearance was obtained from Institutional Review Board (IRB) of College of Medicine and Health Sciences, Bahir Dar University with the approval number of 459/2022. Prior to data collection, informed verbal consent was obtained from each study participants. The informed verbal consent was accepted by IRB since the study has minimal risk. The respondents were given full right to withdraw from the interview whenever they feel uncomfortable. Furthermore, confidentiality was kept by excluding name of the respondents from data collection tool and instead we used unique identification number as a code.

## Results

### Sociodemographic and socioeconomic factors

Four hundred two (402) household heads were interviewed, making a response rate of 95%. From which 69.4% (279) of the households were led by male, the mean and standard deviation of age of household heads were 44.1±14.91 with minimum and maximum value of 20 and 100 respectively. About 40% (161) of the household heads were found to be the age category of 31–45. About 99.5% (400) and 90% (362) of the household heads were Amhara and Orthodox Christian, respectively. From the participants, 10.7% (43) were cannot read and write and 69.2% (278) of them were married. About 75.6% of the households had family size of less than or equal to 4. Regarding wealth status of the households, 19.9%, 20.1%, 19.9%, 21.4%, and 18.7% of the households were fall in first, second, third, fourth and fifth quintiles respectively (Table 1).

### Household annual consumption expenditure

The mean annual household expenditure (food expenditure: ETB47791.34 ($901.72) and non-food expenditure: ETB42033.35 ($793.08)) was ETB89824.69 ($1694.81) with standard deviation of 45826.33($864.65). Whereas, the mean household annual healthcare expenditure was ETB12050.64 ($227.37) with the standard deviation of 25299.87($ 477.36) (Table 2).

**Table 1. Sociodemographic and socioeconomic characteristics of households, Debre Tabor town, Ethiopia, 2022.**

| Variables | Category | Frequency | Percent (%) |
|---|---|---|---|
| Sex of household head | Male | 279 | 69.4 |
| | Female | 123 | 30.6 |
| Age of household head | < = 30 | 92 | 22.9 |
| | 31–45 | 161 | 40.0 |
| | 46–60 | 90 | 22.4 |
| | >60 | 59 | 14.7 |
| Marital status of household head | Married | 278 | 69.2 |
| | Unmarried | 124 | 30.8 |
| Educational status of household head | No education | 43 | 10.7 |
| | Read and write only | 31 | 7.7 |
| | primary(1–8) | 52 | 12.9 |
| | secondary(9–12) | 66 | 16.4 |
| | College and above | 210 | 52.2 |
| Occupation of household head | Self employed | 195 | 48.5 |
| | Government employed | 188 | 46.8 |
| | Private sectors | 19 | 4.7 |
| Family size | < = 4 | 304 | 75.6 |
| | >4 | 98 | 24.4 |
| Presence of U5 Children | Yes | 125 | 31.1 |
| | No | 277 | 68.9 |
| Wealth status | Quintile 1 | 80 | 19.9 |
| | Quintile 2 | 81 | 20.1 |
| | Quintile 3 | 80 | 19.9 |
| | Quintile 4 | 86 | 21.4 |
| | Quintile 5 | 75 | 18.7 |

**Table 2. Annual total expenditure of households, Debre Tabor town, Ethiopia, 2022.**

| HH Annual expenditure | N | Mean (ETB) | Std. Dev | Median |
|---|---|---|---|---|
| Total household expenditure | 402 | 89824.69 | 45826.33 | 80548 |
| Household food expenditure | 402 | 47791.34 | 21061.86 | 43800 |
| Nonfood household expenditure | 402 | 42033.35 | 31141.96 | 33695 |
| Annual direct medical cost | 402 | 5036.97 | 10824.95 | 1096 |
| Registration card, | 402 | 174.02 | 492.463 | 50 |
| Medications | 402 | 2874.71 | 500 | 5980.98 |
| Imaging diagnostic test | 402 | 876.19 | 2434.26 | 0 |
| Laboratory | 402 | 812.24 | 200 | 2832.53 |
| Bed | 402 | 299.81 | 2360.39 | 2832.53 |
| Annual direct nonmedical cost | 402 | 865.1 | 3494.16 | 100 |
| Transport | 402 | 318.36 | 837.16 | 100 |
| Cafeteria | 402 | 481.69 | 3026.45 | 0 |
| Lodging | 402 | 65.05 | 345.1 | 0 |
| Indirect health cost (lost days) | 402 | 5622.1 | 13035.5 | 1996.5 |
| Total Health expenditure | 402 | 12050.64 | 25299.87 | 4120.5 |

NB: All monetary values were explained in Ethiopian birr

N is number of observations and Std. Dev: Standard deviation.

## Health and health related characteristics

One or more household members sought modern healthcare in 83.8% (337) of the households and from these, about 6.2% (27) of the sick members have had referral history. The percentage of households which have at least one chronic health condition was 32.3% and 21.9% of the households sought healthcare from traditional healers (Table 3).

## Financial hardship of healthcare

About 37.1% (149), 11.2% (45) and 15.9% (64) of the households encountered catastrophic health expenditure at 10%, 25% and 40% nonfood threshold levels, respectively. Moreover, 10.4% (42) of the households were pushed below extreme poverty line ($1.90 a day extreme poverty line) because of healthcare expenditure. From participants with a history of referral,

**Table 3. Health and health related characteristics of households, Debre Tabor town, Ethiopia, 2022.**

| Variables | Category | Frequency | Percent (%) |
|---|---|---|---|
| Modern healthcare seek | Yes | 337 | 83.8 |
|  | No | 65 | 16.2 |
| Health institution type | Public | 228 | 56.7 |
|  | Private | 109 | 27.1 |
| Admission history | Yes | 35 | 8.7 |
|  | No | 302 | 75.1 |
| Referral history | Yes | 27 | 6.7 |
|  | No | 310 | 77.1 |
| Chronic health conditions | Yes | 130 | 32.3 |
|  | No | 272 | 67.7 |
| Traditional healthcare seek | Yes | 88 | 21.9 |
|  | No | 314 | 78.1 |

**Table 4. Financial hardship of healthcare among households in Debre Tabor town, Ethiopia, 2022.**

| Variables | Measurements | At 10% threshold | At 25% threshold | At 40% threshold |
|---|---|---|---|---|
| Catastrophic health expenditure | Catastrophic headcount (%) | 37.1 | 11.2 | 15.9 |
| | Catastrophic overshoot | 20.05 | 7.32 | 12.51 |
| | Mean positive gap (%) | 54.04 | 65.36 | 78.68 |
| | Measurements | Prepayment | Post payment | Discrepancy |
| Impoverishing health expenditure | Poverty headcount (%) | 70.4 | 80.8 | 10.4 |
| | Poverty gap | 9527.21 | 11848.68 | 2321.47(24.37%) |
| | Normalized poverty gap | 94.33 | 117.31 | 22.98 |

27, 26(96.35%) of them experienced catastrophic health expenditure which attributes 17.45% of catastrophic households. About ETB9527.21 ($179.76) and ETB11 848.68 ($223.56) were needed to bring the poor households to poverty line before and after healthcare expenditure, respectively. An additional ETB2321.47 ($43.80) was needed to bring the poor households to poverty level after expending for healthcare services (Table 4).

## Coping mechanisms of healthcare expenditure

Among the households, 99% used own savings as a source of fund for healthcare cost. Moreover, 3.7% and 5.5% used selling household asset and borrowing as a coping mechanism for their health expenditure. About 22.6% of the households were found to be insured with community based health insurance (CBHI) (Table 5).

## Factors associated with catastrophic health expenditure

From bi-variable regression, about 15 variables were candidates (p<0.2) for multiple logistic regression: These were sex of household head, age of household head, religion of household head, educational status of household head, occupation of household head, presence of under 5 children (U5C), family size, insurance status, hospitalization, health institution type, presence of chronic health conditions and seeking healthcare from traditional healers. Finally, from multiple logistic regression, age of household head, occupation of household head, insurance status, having social support, having chronic health conditions, sought healthcare from traditional healers were found to be statistically significant (at p<0.05) factors for CHE.

For instance, households with a head of age in the interval between 31 and 45 years old were 2.5 times higher odds (AOR: 2.5, CI: 0.1.071, 5.82) to encounter catastrophic health expenditure than that of the households with a head in the age less than or equal to 30. Moreover, odds of facing CHE among households with a household head of age 60 and above was 4.213 (AOR: 4.213, CI: 1.23, 14.448) as compared to that of the households with a head whose age less than or equal to 30. Furthermore, the odds of catastrophic health expenditure among

**Table 5. Households coping mechanism for healthcare cost among households in Debre Tabor town, Ethiopia, 2022.**

| | Category | Frequency | Percent (%) |
|---|---|---|---|
| Insurance status | Insured | 91 | 22.6 |
| | None insured | 311 | 77.4 |
| Main source of fund for healthcare cost | Own savings | 398 | 99.0 |
| | Social support | 79 | 19.7 |
| | Borrowing | 22 | 5.5 |
| | Selling assets | 15 | 3.7 |

non-insured households was 2.188 (AOR: 2.188, CI: (1.037, 4.619) as compared to that of the insured households.

Additionally, households having members with chronic health conditions were 7.201 times higher odds (AOR: 7.201, CI: 3.64, 14.262) to be experienced catastrophic health expenditure as compared to that of households not having members with chronic health conditions. Likewise, households whose a member seek healthcare from traditional healers were 2.632 times higher odds (AOR: 2.632, CI: 1.372, 5.046) to encounter catastrophic health expenditure as compared to that of the households members not seeking healthcare from traditional healers.

Households which had no social support were 2.773 times higher odds (AOR: 773, CI: 1.246, 6.170) to face catastrophic health expenditure as compared to that of households having social support (Table 6).

## Discussion

This study aimed to assess financial hardship of health care in terms of the incidence of catastrophic health expenditure (CHE) and impoverishing health expenditure (IHE), including the determinants of CHE, among households in Debre Tabor town. The incidence of CHE was 37.1% and the proportion of impoverished households due to health expenditure was 10.4%. This study implies that the financial hardship of health care is stronger among the less privileged populations: the non-insured, the chronically ill, the elder and socially unsupported. Moreover, avoiding impoverished health expenditure can reduce more than one tenth of poor households.

The incidence of CHE in the current study was higher than that of a previous study conducted, 2015/2016, in Ethiopia which was 2.1% [29]. The possible reason might be due to time and our study included indirect medical costs (lost days due to the illness) which were not considered in the previous study. The other probable reason might be due to the fact that the previous study used secondary data (from 2015/16 HCE and WM survey).

Moreover, the incidence of catastrophic health expenditure in this study was higher compared with the studies conducted on CHE and impoverishment in households of persons with depression in 2019 and CHE for households of people with severe mental health disorder (SMD) in 2015 in rural Ethiopia which stated the incidence of CHE, 20% and 20.3% using 10% threshold level, respectively [14]. The probable reason of this discrepancy might be due to the fact that the time of the current study used latest primary data whereas the previous studies were conducted since 2015.The other possible explanation may be escalation of health care cost due to the COVID-19 pandemic while conducting this study.

However, the incidence of catastrophic health expenditure in the current study was lower by half than that of the findings of the study conducted on economic burden of diabetic mellitus healthcare at Bahir Dar public hospitals in 2020 with the incidence of catastrophic health expenditure of 74.3% using the same, 10%, threshold level [12]. The possible explanation for this difference might be due to the fact that the current study included insured households and non-ill household members, which may lower the incidence of the catastrophic health expenditure, that were not included in the previous study. The other possible reason might be the fact that the current study is conducted on households regardless of the diseases status of the members, whereas the previous study was conducted on diabetic patients, that indicates those households with the presence of household member with chronic conditions (e.g. DM) are prone to CHE.

Similarly, the incidence of catastrophic health expenditure in this study was lower than the findings of the study conducted on financial risk of seeking maternal and neonatal healthcare in southern Ethiopia in 2020 (incidence of CHE: 46% at 10% threshold level of total household

**Table 6. Multiple regression of factors affecting catastrophic health expenditure among households in Debre Tabor town, 2022.**

| Variables | Category | CHE | | COR(95%CI) | AOR(95%CI) |
|---|---|---|---|---|---|
| | | No | Yes | | |
| Sex of HH head | Male | 167 | 112 | 1 | 1 |
| | Female | 86 | 37 | 0.642(0.408, 1.010) | 0.790(0.400, 1.560) |
| Age of Household head | < = 30 | 75 | 17 | 1 | 1 |
| | 31–45 | 107 | 54 | 2.226(1.198, 4.238) | **2.5(1.071, 5.821)**\* |
| | 46–60 | 49 | 41 | 3.691(1.888, 7.216) | 1.884(0.725, 4.897) |
| | >60 | 22 | 37 | 7.42(3.321, 15.636) | **4.213(1.23, 14.448)**\* |
| Religion | Orthodox | 224 | 138 | 1 | **1** |
| | Others | 29 | 11 | 0.616(0.298, 1.292) | 0.504(0.182, 1.397) |
| Educational status of HH head | No education | 24 | 19 | 1 | 1 |
| | Read & write only | 20 | 11 | 0.695(0.269, 1.797) | 0.616(0.164, 2.319) |
| | Primary | 37 | 15 | 0.512(0.219, 1.198) | 0.443(0.128, 1.527) |
| | Secondary | 44 | 22 | 0.632(0.287, 1.392) | 0.761(0.220, 2.630) |
| | College and above | 128 | 82 | 0.809(0.417, 1.570) | 0.791(0.202, 3.099) |
| Occupation of HH head | Self employed | 135 | 60 | 1 | 1 |
| | Gov't employed | 108 | 80 | 1.667(1.096, 2.536) | 0.809(0.312, 2.095) |
| | Private sectors | 10 | 9 | 2.025(0.783, 5.239) | **6.344(1.765, 22.80)**\* |
| Presence of U5C | No | 163 | 114 | 1 | 1 |
| | Yes | 90 | 35 | 0.556(0.352, 0.879) | 0.786(0.405, 1.528) |
| Family size | < = 4 | 199 | 105 | 1 | 1 |
| | >4 | 54 | 44 | 0.648(0.0.408, 1.03) | 0.881(0.440, 1.762) |
| Wealth status | Quintile 1 | 56 | 24 | 0.608(0.313, 1.181) | 0.637(0.230, 1.764) |
| | Quintile 2 | 60 | 21 | 0.497(0.252, 0.978) | 0.640(0.243, 1.687) |
| | Quintile 3 | 52 | 28 | 0.764(0.399, 1.464) | 0.961(0.400, 2.308) |
| | Quintile 4 | 41 | 45 | 1.558(0.834, 2.910) | 1.126(0.490, 2.589) |
| | Quintile 5 | 44 | 31 | 1 | 1 |
| Insurance status | Insured | 66 | 25 | 1 | **1** |
| | None insured | 187 | 124 | 1.751(1.048, 2.925) | **2.188(1.037, 4.619)**\* |
| Chronic health conditions | Yes | 36 | 94 | 10.302(6.344, 16.73) | **7.201(3.64, 14.262)**\* |
| | No | 217 | 55 | 1 | 1 |
| Institution type | Public | 127 | 101 | 1 | 1 |
| | Private | 61 | 48 | 0.989(0.625, 1.367) | 1.481(0.793, 2.764) |
| Admission history | Yes | 9 | 26 | 4.204(1.904, 9.282 | 2,571(0,917, 7.209) |
| | No | 179 | 123 | 1 | 1 |
| Traditional healthcare seek | Yes | 42 | 46 | 1 | 1 |
| | No | 211 | 103 | 2.244(1.388, 3.626) | **2.632(1.372, 5.046)**\* |
| Social Support | Yes | 226 | 97 | 1 | 1 |
| | No | 27 | 52 | 4.487(2.662, 7.565) | **2.773(1.246, 6.170)**\* |
| Borrowing | Yes | 27 | 52 | 6.388(2.305, 3.626 | 2.722(0.723, 10, 255) |
| | No | 226 | 97 | 1 | 1 |

\*means significant at p<0.05.

expenditure) [30]. The possible reason might be due to the fact that mothers and neonates need more healthcare services in nature.

The incidence of CHE in this study was higher compared with study conducted at household level in African countries like Kenya, Uganda, Morocco and South Africa which stated

the incidence of CHE(using 10% threshold level) 10.7%, 14.2%, 1.77% and 5%, respectively [8,31–33]. The possible reason this discrepancy might be due to contexts such as sociodemographic and socioeconomic characteristics are not the same.

The incidence of CHE in our study was also higher than that of the global monitoring for financial protection reports of 2019 and 2021 with incidences of CHE 12.7% and 13%, respectively [1,2]. The probable reason behind the difference might be due to the differences on the scope and context of the studies and the global reports mainly relied on the national report which is secondary data.

Moreover, the percentage of the poverty impact of healthcare expenditure in the current study (IHE: 10.4%) was higher than that of similar studies, conducted on households, in national context, in Ethiopia in 2020 with IHE of 0.9% [29], and conducted on diabetic mellitus patients in Bahirdar city public hospitals with IHE of 5% [12], conducted on financial risk of seeking maternal and neonatal care, in southern Ethiopia, with IHE of 0.3% [30] and conducted on patients with depression in Ethiopian rural households with IHE of 5.8% [14]. The probable reasons behind this deference might be due to the fact that the current study included all household members standardized with adult equivalent size in in terms of sex and age where as the previous studies conducted on specific diseases. Moreover, the discrepancy may be the fact that the cost of life, including escalation of health care cost due to the COVID-19 pandemic, at the time of conducting this study is more costly than the previous.

In addition, the IHE in this study was higher than that of the studies conducted in various African countries like Kenya, Uganda, Morocco and South Africa with IHE were 2.2%, 2.7%, 1.11% and 5% [8,31–33], respectively. The possible explanation might be due to the fact that the difference in different poverty lines (e.g. Kenya used its national poverty line), sociodemographic and socioeconomic characteristics and the difference strategies used in Ethiopia and other African countries to protect their citizens from financial risk of seeking essential health services.

Furthermore, households led by heads with age 60 years and above were higher odds to spend catastrophic expenditure. This is supported by evidence in the study conducted on catastrophic health expenditure of SMD in rural Ethiopia and in Kenya in 2018 [30,32]. Likewise, non-insured households were more vulnerable to catastrophic health expenditure. This implied that health insurance is one way to safeguard households from financial risk of healthcare. This was supported by the study conducted in Kenya in 2018, which revealed that households with one member enrolled for health insurance were protected from catastrophic health expenditure [32].

Additionally, presence of chronic health conditions among household members had strong positive association with catastrophic health expenditure. This implied that chronic health conditions are the main source for financial risk for healthcare expenditure. This finding was supported by the evidence in the study conducted in southern Ethiopia, rural Ethiopia and Kenya [12,14,30,32].

The main limitation of this study was recall bias. Although, measures have been taken like triangulating self-reported health expenditure with the recipients, to reduce recall bias, it is still the limitation of this study.

## Conclusion

This study revealed that significant number of households in Debre Tabor town faced catastrophic health expenditure. The financial hardship of health care is stronger among the less privileged populations: the non-insured, the chronically ill, the elder and socially unsupported.

## Recommendations

Based on the key findings, we would like to recommend the following points for the respective concerned bodies

### Health policy makers

Better to design strategies to increase household enrollment to health insurance to pool the financial risks of households.

Better to design strategies that enhance social support among households

Better to design strategies to aware the community about the traditional health care, including their pros and cons.

Better to give high emphasis on financial risk protection of households with elderly members

Better to give emphasis on financial risk protection of households with chronic diseases

### Health care providers

Better to enhance community based health insurance enrolment of households in Debre tabor town

Better to enhance social support among households

### Community

Being enrolled to health insurance

Enhance social support among households

Prefer modern health care to protect themselves from financial hardship

### Researchers (Academicians)

Conduct prospective study to estimate the actual costs(to avoid recall bias) and financial hardship of health care

## Supporting information

**S1 Text. Survey questionnaire (English version).**
(DOCX)

**S2 Text. Survey questionnaire(Amharic version).**
(DOCX)

## Acknowledgments

We would like to thank Bahir Dar University, study participants, data collectors and supervisor for their contributions for the study. We also thank Debre Tabor town administration and South Gondar zone health department for their support during the process of the study.

## Author Contributions

**Conceptualization:** Yawkal Tsega, Gebeyehu Tsega, Getasew Taddesse, Gebremariam Getaneh.

**Data curation:** Yawkal Tsega, Gebeyehu Tsega.

**Formal analysis:** Yawkal Tsega, Gebeyehu Tsega.

**Funding acquisition:** Yawkal Tsega, Gebeyehu Tsega.

**Investigation:** Yawkal Tsega, Gebeyehu Tsega, Gebremariam Getaneh.

**Methodology:** Yawkal Tsega, Gebeyehu Tsega, Getasew Taddesse.

**Project administration:** Yawkal Tsega, Gebeyehu Tsega, Gebremariam Getaneh.

**Resources:** Yawkal Tsega, Gebeyehu Tsega.

**Software:** Yawkal Tsega, Gebeyehu Tsega.

**Supervision:** Yawkal Tsega, Gebeyehu Tsega, Getasew Taddesse, Gebremariam Getaneh.

**Validation:** Yawkal Tsega, Gebeyehu Tsega, Gebremariam Getaneh.

**Visualization:** Yawkal Tsega, Gebeyehu Tsega.

**Writing – original draft:** Yawkal Tsega, Gebeyehu Tsega.

**Writing – review & editing:** Yawkal Tsega, Gebeyehu Tsega, Getasew Taddesse, Gebremariam Getaneh.

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
