## [Decision Letter · Decision Letter 0]

1 Nov 2022

PONE-D-22-22796Leaving no one behind in health: financial hardship to access health care in EthiopiaPLOS ONE

Dear Dr. Tsega,

Thank you for submitting your manuscript to PLOS ONE. After careful consideration, we feel that it has merit but does not fully meet PLOS ONE’s publication criteria as it currently stands. Therefore, we invite you to submit a revised version of the manuscript that addresses the points raised during the review process.

ACADEMIC EDITOR:Please address my following comments along with reviewers' comments:1. It is multiple logistic regression, not multivariate logistic regression. So, please change it throughout the manuscript.2. Before modelling factors using multiple logistic regression you need to develop a conceptual framework of your own or use an established framework from the literature. This conceptual model will serve as hypotheses that will guide your modelling process. For example, you may find some factors not statistically significant but they could be theoretically important for the Ethiopian contexts and you need to retain those factors in your final parsimonious model although they are not statistically significant.3. Your interpretation of OR is wrong. You interpreted OR like RR. OR 2.5 means 2.5 times higher odds, not 2.5 times more likely!

We look forward to receiving your revised manuscript.

Kind regards,

Khurshid Alam, Ph. D.

Academic Editor

PLOS ONE

Journal Requirements:

2. Please provide additional details regarding participant consent. In the ethics statement in the Methods and online submission information, please ensure that you have specified whether: 1) whether the ethics committee approved the verbal/oral consent procedure, 2) why written consent could not be obtained, and 3) how verbal/oral consent was recorded. If your study included minors, please state whether you obtained consent from parents or guardians in these cases. If the need for consent was waived by the ethics committee, please include this information.

“no”

“NO authors have competing interests”

Reviewers' comments:

Reviewer's Responses to Questions

**Comments to the Author**

1. Is the manuscript technically sound, and do the data support the conclusions?

Reviewer #1: Yes

Reviewer #2: Yes

2. Has the statistical analysis been performed appropriately and rigorously? 

Reviewer #1: Yes

Reviewer #2: Yes

3. Have the authors made all data underlying the findings in their manuscript fully available?

Reviewer #1: Yes

Reviewer #2: Yes

4. Is the manuscript presented in an intelligible fashion and written in standard English?

Reviewer #1: Yes

Reviewer #2: Yes

5. Review Comments to the Author

Reviewer #1: Comments

The paper is about “Leaving no one behind in health: financial hardship to access health care in Ethiopia” which is very important and interesting research area for Ethiopia and beyond, as it is clearly shown in the introduction part of the manuscript that financial hardship (risk) protection is the international and national agenda, such as SDGs and Health Sector Transformational Plan two(HSTPII), which is a five year(2021-2025) plan (part of the long Ethiopia’s health sector’s fifteen years strategic plan (2015-2035)).As per the findings of this study, due to financial hardship, the poor people(found in low income countries including Ethiopia) either forgone and/or delayed for accessing health care they need or they are obligated to access the needy health care on the expense of other basic needs like food which is against the zero hunger goal of the agenda 2030. In the era of Covid-19 pandemic and other disasters (e.g. conflict and climate change), the poor left behind in health (care).The findings of this study indicated that financial hardship becomes a significant barrier to achieve the notion of leave no behind. For example, impoverished health care makes, about 10.4%, households became poor and catastrophic health expenditure was found in 37.1% of the households.

Hence, it is very relevant and suitable to publish this evidence that showed the extent of financial hardship (and the associated factors) on households in Ethiopia in general and in Debre Tabor town (one of conflict affected areas of Ethiopia) in particular. This evidence will inform policy options in the country in order to attain the universal health coverage (leave no one behind in health).Despite, there are few comments that need to be either clear or addressed before this paper could be published.

1. The threshold that the authors used for catastrophic health expenditure is 10%. However, they put the results in three thresholds (10%, 25% and 40% non-food). Why this is so?

2. In the introduction and discussion sections of the manuscript, there are long sentences that can be reconstructed to simple, short and concise sentences as per the plos one journal recommendation.

3. The authors should convert the Ethiopia Birr to US Dollar in bracket to make it clear for international community.

4. in the methods section your dependent variable says Financial hardship of healthcare(CHE ) : here the abbreviation CHE is not represent financial hard ships of health care so use either of the two / catastrophic health expenditure or financial hard ship

5. In the measurement of catastrophic health expenditure as you clearly present the threshold ranges from 10%- 40 %. So why you select the lower value (10%)? Because of this decision you may under estimate the level of catastrophic health expenditure in the study area? Say something about this issue in your manuscript.

Reviewer #2: Really the result was technically sound and It is so in my professional experience. As I am the CEO of the private Hospital in Bahir Dar, nearby city and the regional capital, I repeatedly saw many customers unable to cover the out of pocket expenditures.

I recommend to study how the user fee setting is set both in public and private health institutions.

Thank you very much.

6. PLOS authors have the option to publish the peer review history of their article (what does this mean?). If published, this will include your full peer review and any attached files.

Reviewer #1: No

Reviewer #2: **Yes: **Habitamu Ayehualem Bayu

---

## [Author Response · Author response to Decision Letter 0]

25 Nov 2022

Response to editor:

We really thank you, editor, for your insightful comments that are vital to improve the manuscript. We accept all of your comments and we corrected accordingly. 

Here, below, are the responses for the comments: 

1. “It is multiple logistic regression, not multivariate logistic regression. So, please change it throughout the manuscript”.

• Yes, you are quite right, we changed to multiple logistic regression throughout the manuscript.

2. “Before modelling factors using multiple logistic regression you need to develop a conceptual framework of your own or use an established framework from the literature. This conceptual model will serve as hypotheses that will guide your modelling process. For example, you may find some factors not statistically significant but they could be theoretically important for the Ethiopian contexts and you need to retain those factors in your final parsimonious model although they are not statistically significant”.

• We have developed the conceptual framework, but not reported in the 1st version of the manuscript. Now, we have reported in the manuscript. 

3. “Your interpretation of OR is wrong. You interpreted OR like RR. OR 2.5 means 2.5 times higher odds, not 2.5 times more likely!”

• Yes, you are correct, in the current version, we have corrected accordingly. 

 Response to reviewers

 Thank you, reviewers, for your constructive comments that are vital to improve the manuscript. We accept all of your comments and we corrected accordingly. 

Here, below, are the responses for the comments and questions raised: 

1. “The threshold that the authors used for catastrophic health expenditure is 10%. However, they put the results in three thresholds (10%, 25% and 40% non-food). Why this is so?”

• Most authors, including in World health organization and World Bank joint report on UHC, used 10% and 25% thresholds. The 40% non-food also used by authors as well, as stated in the methods part of the manuscript. Therefore, we used the 10% threshold for statistical modeling which is the WHO threshold. However, we stated (in the manuscript) that the incidence CHE with the three thresholds (10%, 25% and 40% non-food) to allow the readers and/or reviewers to compare the incidence with these three thresholds. This will give more information than using only 10% threshold. 

2. “In the introduction and discussion sections of the manuscript, there are long sentences that can be reconstructed to simple, short and concise sentences as per the plos one journal recommendation”.

• We have reconstructed accordingly.

3. “The authors should convert the Ethiopia Birr to US Dollar in bracket to make it clear for international community”

• We have converted in to $ in bracket as per the comment.

4. “In the methods section your dependent variable says Financial hardship of healthcare (CHE): here the abbreviation CHE is not represent financial hardships of health care so use either of the two / catastrophic health expenditure or financial hard ship”.

• We corrected as per the comment 

5. “In the measurement of catastrophic health expenditure as you clearly present the threshold ranges from 10%- 40 %. So why you select the lower value (10%)? Because of this decision you may under estimate the level of catastrophic health expenditure in the study area?”

• This is because 10% the usual threshold that used in global reports, including WHO and WB. Hence, the audience can compare the findings with that of such reports (e.g WHO reports on UHC). However, we have reported the incidence/headcount at 25% and 40% non-food as well to give additional information at these thresholds.

---

## [Editor Report · Decision Letter 1]

21 Feb 2023

Leaving no one behind in health: financial hardship to access health care in Ethiopia

PONE-D-22-22796R1

Dear Dr. Tsega,

We’re pleased to inform you that your manuscript has been judged scientifically suitable for publication and will be formally accepted for publication once it meets all outstanding technical requirements.

Kind regards,

Khurshid Alam, Ph. D.

Academic Editor

PLOS ONE
---

## [Editor Report · Acceptance letter]

2 Mar 2023

PONE-D-22-22796R1 

Leaving no one behind in health: financial hardship to access health care in Ethiopia 

Dear Dr. Tsega:

I'm pleased to inform you that your manuscript has been deemed suitable for publication in PLOS ONE. Congratulations! Your manuscript is now with our production department. 

Kind regards, 

on behalf of

Dr. Khurshid Alam 

Academic Editor

PLOS ONE